# Analysis of the Psychometric Properties of the Motivation and Strategies of Learning Questionnaire—Short Form (MSLQ-SF) in Spanish Higher Education Students

**Felix Zurita Ortega** [1,]*, **Asuncion Martinez Martinez** [2], **Ramon Chacon Cuberos** [2]
**and Jose Luis Ubago Jiménez** [1]

1.  Department of Didactics of Musical, Plastic and Corporal Expression, University of Granada, 18071 Granada, Spain; jlubago@ugr.es
2.  Department of Research and Diagnosis Methods in Education, University of Granada, 18071 Granada, Spain; asuncionmm@ugr.es (A.M.M.); rchacon@ugr.es (R.C.C.)
*   Correspondence: felixzo@ugr.es; Tel.: +34-958-248-949

**Abstract:** Background and methods: The aim of this research was to analyze the psychometric properties of the Motivation and Learning Strategies Questionnaire-Short Form (MSLQ-SF), using exploratory techniques with university students. The sample was formed by 597 participants aged between 19 and 28 years old (M = 23.04; SD = 3.71), with 156 (26.1%) being male and 441 (73.9%) being female. The exploratory factor analysis was conducted using the FACTOR program. Results: The results indicate that the questionnaire provides high reliability indexes to α = 0.70 for all included dimensions. The factor describing intrinsic orientation towards goal setting was removed following exploratory analysis, while other factors adjusted satisfactorily. All factors were correlated directly and positively ($p < 0.01$). Conclusions: It can be concluded that the MSLQ-SF fulfils the validity and reliability specifications for use with university students of social sciences and health sciences.

**Keywords:** learning; motivation; higher education; students

## 1. Introduction

Interest in the quality of learning in university education has increased in the last few years (Romero et al. 2010). Better conditions have been provided for the investigation of the learning process, since the pupil must open a strategies series to learn, to develop goals, to distribute resources, and to maintain their motivation. A main aim of higher education is to prepare the individual for transition into the world of work. Similarly to other stages of education, self-regulation is seen as an important prerequisite of quality learning, and this is indicated in current theories of learning (Lee et al. 1993). Although, as many authors have indicated, the hypothesis that higher educational level increases the production level of a country is not fulfilled (Neycheva 2016), it is true that resources reverberate in a positive way in the student learning processes.

Diverse investigations exist about the study of motivation as an essential element to understand academic performance in the university field (Gandomkar et al. 2016; Oh et al. 2016), and there is a consistent clear interest to know the diverse factors that can affect this process. If we start from the components that make up metacognitive learning, this indicates that the student should be able to check and mediate between their own personal intelligence and capacity. It is argued that a student always promotes and improves his/her motivation and learning strategies, particularly when he/she obtains an effective personal learning environment (Kassab et al. 2015). Motivation is one of the most

studied factors in psychology, due to its great potential to explain human behavior, as motivation influences certain actions, modifying their intensity and direction (Castro-Sánchez et al. 2018).

Various studies have revealed the relationship between learning, motivation, and teaching (Hall et al. 2016; Thomas and Muller 2016), and there are many investigations that have focused on studies in the higher education field, as indicated (Angeli dos Santos and Ferreira 2016; Froiland and Worrell 2016). Among the characteristics that make up this instrument is the evaluation of self-regulatory motivation and learning strategies, as well as management of the learning context by university students. The ability to learn has been established as one of the main objectives of university education. Strategies for learning have been associated with intercultural transferability (Tong et al. 2019), sports (González-Valero et al. 2017; Castro-Sánchez et al. 2019), technologies (Virtanen et al. 2018), or job insertion (Martínez-Martínez et al. 2017), among others.

Therefore, it is necessary to be provided with instruments and technical tools that are based on empirical evidence and are capable of extending knowledge and informing interventions in the context of higher education. Various tools in the present line of research have been validated for use in specific contexts around the world. For instance, Castañeda (2004) and Sabogal et al. (2011) have used the Inventory on Learning Styles and Motivational Orientation (EDAOM, according to its initials in Spanish) and produced Cronbach alpha values greater than 0.900. Pintrich (1988) applied the Motivational Strategies for Learning Questionnaire (MSLQ) and produced a reliability value of 0.750. Pintrich et al. (1993) later modified this same instrument, reducing it to 40 items and naming it Motivation and Learning Strategies Questionnaire-Short Form (MSLQ-SF), with Roces et al. (1995) then adapting it into Spanish and renaming it CEAM II (Learning and Motivation Strategies Questionnaire). Additionally, it was used by Riveiro et al. (2005) in an educational sciences sample, using only the motivational and strategies scale. Few studies have focused on the evaluation of the psychometric properties of the instrument at the Spanish-speaking level, specifically Sabogal et al. (2011) in Colombia, Ramírez et al. (2013) in Mexico, or more recently, Inzunza et al. (2018) in Chile.

In the global context (Venezuela, Czech Republic, or Malaysia), some authors, such as Rao and Sachs (1999), Jakesöval and Hrbackova (2014), or Abdullah (2016), carried out studies in primary and secondary education populations, looking for knowledge on performance, motivation, and learning strategies. Recently, Angeli dos Santos and Ferreira (2016) analyzed the EMAPRE-U reliability and its validity in Portuguese populations, and Thomas and Muller (2016) used the SMR-L in Austria. Furthermore, Hwang et al. (2019) adopted the Online Technologies Self-Efficacy Scale (OTSES).

The link between motivation and emotions and learning is closely related. However, as posed by Pekrun (2006) and Hall et al. (2016), there are very scarce gauges to explore this relationship. Not much information has been found related to the motivation, learning, and resources in higher education. Accordingly, this area that the present paper focuses on is a fertile field for research, given that motivation intervenes in a direct way in learning dimensions.

Therefore, it appears that the nine-factor and three-dimensional model has not been sufficiently tested through exploratory techniques with university students. The aim of the study was to analyze the psychometric properties of the Motivational Strategies for Learning Questionnaire-Short Form (MSLQ-SF). It aimed to calculate the reliability of the instrument for the purpose of adaptation and application to the Spanish university population. For this purpose, an Exploratory Factorial Analysis (AFE) was carried out.

## 2. Materials and Methods

### 2.1. Participants

The sample was formed of 597 university students aged between 19 and 28 years old (M = 23.04 years; SD = 3719), with $n = 156$ being male (26.1%) and $n = 441$ being female (73.9%). The sample was stratified according to three cities where the participants were attending university: 103 in Granada (17.3%), 138 in Ceuta (23.1%), and 356 in Melilla (59.6%). With regards to the area being

studied, 280 students in the sample were studying social sciences (46.9%) and 317 were studying health sciences (53.1%). The sample was obtained using simple random sampling. It is necessary to indicate that 28 questionnaires were excluded due to being incorrect or incomplete. The sample distribution and the sociodemographic data can be seen in Table 1.

**Table 1.** Sample distribution and demographic data.

| University Campus | Granada | Ceuta | Melilla | TOTAL |
|---|---|---|---|---|
| **Male** | 12.2% (*N* = 19) | 23.7% (*N* = 37) | 64.1% (*N* = 100) | 156 (100.0%) |
| **Female** | 19.0% (*N* = 84) | 22.9% (*N* = 101) | 58.0% (*N* = 256) | 441 (100.0%) |
| **Knowledge Area** | **Social Science** | | **Health Science** | |
| **Male** | 54.5% (*N* = 85) | | 45.5% (*N* = 71) | 156 (100.0%) |
| **Female** | 44.2% (*N* = 195) | | 55.8% (*N* = 246) | 441 (100.0%) |

### 2.2. Instruments

The present study used the Motivational Strategies for Learning Questionnaire-Short Form (MSLQ-SF) developed by Pintrich et al. (1993). This questionnaire consists of 40 questions grouped according to three dimensions:

(a) Motivation scale: composed of items relating to task evaluation (Items 20, 26, and 39) and anxiety responses (Items 3, 12, 21, and 29).
(b) Learning strategies: formed by developmental strategies (Items 4, 5, 22, 24, and 25), organizational strategies (Items 13, 14, 23, and 40), critical thinking (Items 1.6 and 15), and self-regulation (Items 16, 30, 31, 32, 34, 35, and 36).
(c) Resource management strategies: structured according to time and study habits (Items 2.8, 17, 18, 33, and 38), effort towards self-regulation (Items 7, 9, 11, 19, 27, and 28), and guidance towards setting intrinsic goals (Items 10 and 37).

Participants responded on a Likert scale, which ranged between 1 and 5, where 1 = "never" and 5 = "always". Overall dimension scores were determined by summing the relevant items provided above.

### 2.3. Procedure

The questionnaires were administered outside of school hours by trained researchers who assisted the students with any queries. Students were assured that their responses would be kept anonymous and that all information provided during the course of the study would be used only for scientific purposes. Participants were not informed of the study aim in order to avoid insincere responses and to minimize as much as possible the effects of social desirability.

### 2.4. Data Analysis

In the present study, the psychometric properties (reliability and exploratory factorial analysis) of the measures were analyzed using the statistical software programs SPSS 22.0 for Windows and FACTOR Analysis 9.3.1 (Lorenzo-Seva and Ferrando 2006). Firstly, SPSS 22.0 was used to analyze the metric properties of each item through basic descriptive statistics (media, dispersion, kurtosis, and asymmetry). Secondly, FACTOR was used to examine the goodness of fit of the data, which is essential when establishing scale validity. This assessment was based on a number of recommended criteria (Bentler 1990; McDonald and Marsh 1990). When conducting exploratory factor analysis, a polychoric matrix was used (Freiberg et al. 2013) alongside the minimum rank factor analysis extraction method (Ten Berge and Kiers 1991). Following the assumption that the included factors were moderately correlated (Worthington and Wittaker 2006), a direct oblimin rotation was also used. Goodness-of-fit indices such as the Kaiser-Meyer-Olkin (*KMO*) index that measures sample adequacy, the Comparative Fit Index (*CFI*), the Goodness of Fit Index (*GFI*), and

the Root Mean Square Error of Approximation (*RMSEA*) were all used. Acceptable values for the *KMO* index were found between 0.05 and 1 as indicated by Suárez (2007). The CFI and GFI indices are considered to show appropriate adjustment of the model when values are higher than 0.90 (Schumacher and Lomax 1996). With regards to the *RMSEA*, a reasonable adjustment is considered to be indicated by values lower than 0.08 (Browne and Cudeck 1993). Cronbach alpha coefficients were used to determine the internal consistency of the instrument and its different dimensions, with values higher than 0.700 being acceptable.

## 3. Results

In the first step of data analysis, descriptive statistics (Table 2) were calculated following the steps recommended by experts (Schmider et al. 2010). The decision was made to remove Variable (V) 08, as it presented figures greater than 2.00 in the dispersion tests (asymmetry and kurtosis).

**Table 2.** Descriptive statistics for items of the Motivation and Learning Strategies Questionnaire-Short Form (MSLQ-SF).

|       | Media | DS    | Variance | Asymmetry | Kurtosis | Range |
|-------|-------|-------|----------|-----------|----------|-------|
| V 01  | 3.51  | 1.333 | 1.777    | −0.529    | −0.894   | 4     |
| V 02  | 3.38  | 1.306 | 1.705    | −0.287    | −1.026   | 4     |
| V 03  | 2.55  | 1.472 | 2.167    | 0.377     | −1.278   | 4     |
| V 04  | 3.87  | 1.172 | 1.373    | −0.724    | −0.491   | 4     |
| V 05  | 4.24  | 1.185 | 1.404    | −1.456    | 0.981    | 4     |
| V 06  | 3.65  | 1.270 | 1.614    | −0.587    | −0.750   | 4     |
| V 07  | 4.37  | 1.050 | 1.102    | −1.606    | 1.546    | 4     |
| V 08  | 4.47  | 0.992 | 0.984    | −1.938    | 2.973    | 4     |
| V 09  | 4.10  | 1.129 | 1.275    | −1.047    | 0.074    | 3     |
| V 10  | 4.32  | 1.051 | 1.104    | −1.547    | 1.598    | 4     |
| V 11  | 3.93  | 1.125 | 1.266    | −0.797    | −0.231   | 4     |
| V 12  | 3.91  | 1.242 | 1.542    | −0.957    | −0.138   | 4     |
| V 13  | 4.07  | 1.243 | 1.544    | −1.168    | 0.229    | 4     |
| V 14  | 4.14  | 1.189 | 1.413    | −1.208    | 0.315    | 4     |
| V 15  | 3.88  | 1.220 | 1.488    | −0.924    | −0.112   | 4     |
| V 16  | 3.82  | 1.142 | 1.305    | −0.793    | −0.183   | 4     |
| V 17  | 3.58  | 1.385 | 1.919    | −0.555    | −0.989   | 4     |
| V 18  | 3.54  | 1.378 | 1.900    | −0.587    | −0.887   | 4     |
| V 19  | 4.22  | 1.058 | 1.119    | −1.276    | 0.797    | 3     |
| V 20  | 2.45  | 1.470 | 2.161    | 0.511     | −1.184   | 4     |
| V 21  | 3.51  | 1.447 | 2.093    | −0.524    | −1.098   | 4     |
| V 22  | 3.85  | 1.179 | 1.391    | −0.779    | −0.357   | 4     |
| V 23  | 3.81  | 1.293 | 1.673    | −0.777    | −0.556   | 4     |
| V 24  | 4.07  | 1.103 | 1.216    | −1.004    | 0.129    | 4     |
| V 25  | 4.08  | 1.127 | 1.271    | −1.079    | 0.262    | 4     |
| V 26  | 2.63  | 1.460 | 2.132    | 0.300     | −1.312   | 4     |
| V 27  | 3.60  | 1.304 | 1.701    | −0.550    | −0.841   | 4     |
| V 28  | 4.06  | 1.117 | 1.248    | −1.068    | 0.244    | 4     |
| V 29  | 3.22  | 1.557 | 2.426    | −0.227    | −1.455   | 4     |
| V 30  | 3.54  | 1.251 | 1.564    | −0.501    | −0.774   | 4     |
| V 31  | 3.48  | 1.223 | 1.495    | −0.374    | −0.789   | 4     |
| V 32  | 3.61  | 1.242 | 1.543    | −0.554    | −0.702   | 4     |
| V 33  | 4.22  | 1.168 | 1.365    | −1.456    | 1.097    | 4     |
| V 34  | 3.97  | 1.145 | 1.311    | −0.930    | −0.016   | 4     |
| V 35  | 3.88  | 1.177 | 1.384    | −0.841    | −0.229   | 4     |
| V 36  | 3.56  | 1.246 | 1.552    | −0.469    | −0.829   | 4     |
| V 37  | 3.85  | 1.151 | 1.325    | −0.775    | −0.276   | 4     |
| V 38  | 3.53  | 1.251 | 1.565    | −0.477    | −0.789   | 4     |
| V 39  | 2.06  | 1.280 | 1.639    | 0.915     | −0.422   | 4     |
| V 40  | 3.80  | 1.345 | 1.809    | −0.802    | −0.620   | 4     |

Following this, the FACTOR Analysis program (Lorenzo-Seva and Ferrando 2006) was used with nine rotating factors entered. Bartlett's test, [12640.7 (df = 780; *p* = 0.000)], and the Kaiser-Meyer-Olkin (KMO) test (=0.955) were used to examine whether the sample came from populations with the same variance and whether the model presented a good match to the empirical data. The nine extracted factors explained 64% of the total variance; the CFI value was 0.96; the GFI was 0.99; the AGFI also obtained 0.99; and the Root Mean Squared Residual (RMSR) value was 0.019. All of these data indicated an excellent fit to the model. The assessment was based on several criteria as recommended by Bentler (1990) and McDonald and Marsh (1990).

Four variables (V 10, V 27, V 38, and V 40) have been removed from Table 3 due to the fact that loading differences on these factors were less than 0.100. The final scale was formed by eight factors (Factor 1 was removed due to being formed by a single variable). The second factor was constituted by seven variables relating to self-regulation. A third factor was comprised of five variables all of which described the effort of self-regulation. A fourth factor related to anxiety responses and was formed by four variables. A fifth dimension was related to evaluation of the task and constituted three variables. A sixth factor described time and study habits and was formed by five variables. The seventh factor referred to organizational strategies and was formed by three variables. The eighth factor described critical thinking and was formed by three items. Finally, the ninth factor was constituted by five variables that corresponded to developmental strategies (Table 3).

**Table 3.** Rotated factor matrix.

|       | F1     | F2     | F3     | F4     | F5     | F6     | F7     | F8     | F9     |
|-------|--------|--------|--------|--------|--------|--------|--------|--------|--------|
| V 01  | 0.152  | 0.127  | 0.116  | 0.174  | −0.040 | 0.175  | 0.144  | **0.494** | −0.182 |
| V 02  | 0.281  | 0.272  | 0.006  | −0.052 | −0.072 | **0.402** | −0.029 | −0.027 | −0.004 |
| V 03  | 0.190  | −0.037 | −0.257 | **0.420** | 0.152  | −0.240 | 0.158  | 0.291  | −0.084 |
| V 04  | 0.248  | 0.412  | 0.124  | −0.097 | −0.077 | −0.197 | 0.078  | −0.030 | **0.477** |
| V 05  | 0.074  | 0.032  | 0.101  | −0.079 | 0.041  | 0.018  | 0.025  | 0.237  | **0.463** |
| V 06  | 0.239  | −0.015 | −0.009 | −0.156 | −0.053 | −0.044 | 0.068  | **0.463** | 0.315  |
| V 07  | 0.033  | 0.145  | **0.468** | −0.029 | −0.009 | −0.249 | 0.162  | −0.003 | 0.314  |
| V 09  | 0.241  | −0.053 | **0.554** | 0.279  | 0.067  | −0.014 | 0.005  | −0.063 | −0.025 |
| V 11  | −0.010 | −0.063 | **0.619** | 0.221  | −0.017 | 0.009  | 0.322  | −0.191 | −0.077 |
| V 12  | 0.060  | −0.273 | 0.064  | **0.425** | 0.238  | −0.114 | 0.199  | 0.249  | 0.110  |
| V 13  | −0.090 | −0.026 | 0.164  | −0.344 | −0.004 | 0.116  | **0.689** | 0.425  | −0.048 |
| V 14  | 0.044  | 0.044  | 0.174  | −0.342 | −0.031 | 0.085  | **0.623** | 0.210  | 0.137  |
| V 15  | 0.042  | 0.172  | −0.167 | −0.155 | −0.006 | 0.233  | 0.078  | **0.302** | 0.232  |
| V 16  | −0.010 | **0.952** | −0.056 | 0.038  | 0.052  | 0.028  | 0.245  | −0.389 | 0.004  |
| V 17  | −0.018 | −0.008 | −0.125 | 0.134  | −0.004 | **0.596** | −0.104 | 0.122  | 0.014  |
| V 18  | 0.053  | −0.015 | −0.156 | 0.301  | −0.085 | **0.526** | −0.234 | 0.164  | −0.070 |
| V 19  | −0.035 | 0.001  | **0.393** | 0.280  | 0.033  | 0.207  | 0.165  | −0.153 | 0.093  |
| V 20  | 0.054  | 0.114  | 0.075  | 0.259  | **−0.431** | 0.041  | 0.087  | 0.173  | −0.134 |
| V 21  | 0.036  | −0.027 | 0.061  | **0.890** | −0.026 | 0.025  | 0.010  | −0.068 | −0.015 |
| V 22  | 0.090  | 0.084  | −0.129 | −0.022 | 0.049  | 0.269  | −0.000 | −0.040 | **0.604** |
| V 23  | −0.029 | −0.027 | 0.079  | −0.091 | 0.014  | 0.103  | **0.444** | 0.393  | −0.147 |
| V 24  | −0.002 | 0.074  | −0.106 | −0.039 | 0.021  | 0.036  | −0.052 | 0.070  | **0.787** |
| V 25  | −0.076 | −0.076 | 0.010  | 0.173  | 0.032  | −0.067 | −0.096 | 0.014  | **0.810** |
| V 26  | −0.008 | 0.274  | 0.054  | 0.171  | **−0.383** | −0.273 | 0.233  | 0.235  | −0.056 |

**Table 3.** *Cont*.

|  | F1 | F2 | F3 | F4 | F5 | F6 | F7 | F8 | F9 |
|---|---|---|---|---|---|---|---|---|---|
| V 29 | −0.053 | 0.070 | −0.006 | **0.884** | −0.161 | 0.047 | −0.090 | −0.049 | 0.001 |
| V 30 | 0.037 | **0.635** | 0.001 | −0.010 | 0.014 | −0.002 | −0.138 | 0.021 | 0.044 |
| V 31 | 0.018 | **1.109** | −0.090 | −0.188 | 0.095 | −0.006 | −0.003 | −0.137 | −0.113 |
| V 32 | −0.018 | **0.735** | −0.020 | −0.039 | 0.029 | −0.053 | −0.099 | 0.044 | 0.091 |
| V 33 | 0.035 | 0.063 | 0.235 | 0.088 | 0.006 | **0.581** | −0.040 | 0.087 | −0.309 |
| V 34 | −0.030 | **0.473** | 0.255 | 0.364 | −0.101 | −0.282 | 0.075 | 0.068 | 0.016 |
| V 35 | −0.174 | **0.521** | 0.030 | 0.116 | −0.066 | 0.242 | −0.006 | −0.103 | −0.001 |
| V 36 | −0.153 | **0.468** | −0.066 | 0.257 | −0.048 | 0.225 | −0.033 | 0.011 | −0.069 |
| V 37 | **0.346** | −0.095 | 0.057 | 0.215 | −0.019 | 0.101 | −0.069 | 0.097 | 0.038 |
| V 39 | −0.048 | 0.015 | −0.240 | 0.118 | **0.376** | 0.080 | 0.067 | 0.009 | −0.360 |

In bold are the values considered.

The following shows the good fit of the questionnaire configuration (Table 4). The reliability coefficient of the questionnaire was 0.938. Following removal of Factor 1 due to it being described by only a single item, all variables obtained a Cronbach alpha with a value of $\alpha > 0.700$. With regards to the various dimensions, acceptable alphas were obtained for the motivation scale ($\alpha = 0.779$), learning strategies ($\alpha = 0.935$), and resource management strategies ($\alpha = 0.878$).

**Table 4.** Factor loading of MSLQ-SF dimensions.

|  | F1 | F2 | F3 | F4 | F5 | F6 | F7 | F8 | F9 |
|---|---|---|---|---|---|---|---|---|---|
| V 37 | **0.346** |  |  |  |  |  |  |  |  |
| V 16 |  | **0.952** |  |  |  |  |  |  |  |
| V 30 |  | **0.635** |  |  |  |  |  |  |  |
| V 31 |  | **1.109** |  |  |  |  |  |  |  |
| V 32 |  | **0.735** |  |  |  |  |  |  |  |
| V 34 |  | **0.473** |  |  |  |  |  |  |  |
| V 35 |  | **0.521** |  |  |  |  |  |  |  |
| V 36 |  | **0.468** |  |  |  |  |  |  |  |
| V 07 |  |  | **0.468** |  |  |  |  |  |  |
| V 09 |  |  | **0.554** |  |  |  |  |  |  |
| V 11 |  |  | **0.619** |  |  |  |  |  |  |
| V 19 |  |  | **0.393** |  |  |  |  |  |  |
| V 28 |  |  | **0.475** |  |  |  |  |  |  |
| V 03 |  |  |  | **0.420** |  |  |  |  |  |
| V 12 |  |  |  | **0.425** |  |  |  |  |  |
| V 21 |  |  |  | **0.890** |  |  |  |  |  |
| V 29 |  |  |  | **0.884** |  |  |  |  |  |
| V 20 |  |  |  |  | **−0.431** |  |  |  |  |
| V 26 |  |  |  |  | **−0.383** |  |  |  |  |
| V 39 |  |  |  |  | **0.376** |  |  |  |  |
| V 02 |  |  |  |  |  | **0.402** |  |  |  |
| V 17 |  |  |  |  |  | **0.596** |  |  |  |
| V 18 |  |  |  |  |  | **0.526** |  |  |  |
| V 33 |  |  |  |  |  | **0.581** |  |  |  |
| V 13 |  |  |  |  |  |  | **0.689** |  |  |
| V 14 |  |  |  |  |  |  | **0.623** |  |  |
| V 23 |  |  |  |  |  |  | **0.444** |  |  |

**Table 4.** *Cont.*

| | F1 | F2 | F3 | F4 | F5 | F6 | F7 | F8 | F9 |
|---|---|---|---|---|---|---|---|---|---|
| V 01 | | | | | | | | 0.494 | |
| V 06 | | | | | | | | 0.463 | |
| V 15 | | | | | | | | 0.302 | |
| V 04 | | | | | | | | | 0.477 |
| V 05 | | | | | | | | | 0.463 |
| V 22 | | | | | | | | | 0.604 |
| V 24 | | | | | | | | | 0.787 |
| V 25 | | | | | | | | | 0.810 |
| α = 0.938 | - | α = 0.867 | α = 0.822 | α = 0.718 | α = 0.769 | α = 0.775 | α = 0.824 | α = 0.707 | α = 0.846 |

As can be observed from Table 5, the Cronbach alpha results were satisfactory both overall and for each of the factors and dimensions, with values of $p > 0.700$ being produced. Following examination of the Pearson correlation coefficients, it can be seen that almost all the factors presented a significant correlation with 99% certainty ($p < 0.01$). Results for Factors 2, 3, 6, and 9 stood out in particular, as values greater than 0.700 were observed. In addition to Factors 3, 6, and 9, Factor 6 related to Factor 7 and Factor 8 alongside Factor 9 are highlighted. All correlations were found to be positive and direct with a significance level of 0.01.

**Table 5.** Correlation of factors.

| | Factor 2 | Factor 3 | Factor 4 | Factor 5 | Factor 6 | Factor 7 | Factor 8 | Factor 9 |
|---|---|---|---|---|---|---|---|---|
| **Factor 2:** Self-regulation | 1 | | | | | | | |
| **Factor 3:** Effort of self-regulation | 0.740 ** | | | | | | | |
| **Factor 4:** Anxiety responses | 0.247 ** | 0.305 ** | | | | | | |
| **Factor 5:** Evaluation of task | 0.030 | −0.080 | 0.294 ** | 1 | | | | |
| **Factor 6:** Time and learning habits | 0.730 ** | 0.723 ** | 0.250 ** | −0.082 * | 1 | | | |
| **Factor 7:** Organizational strategies | 0.648 ** | 0.650 ** | 0.271 ** | −0.003 | 0.697 ** | 1 | | |
| **Factor 8:** Critical thinking | 0.673 ** | 0.639 ** | 0.269 ** | 0.085 * | 0.657 ** | 0.629 ** | 1 | |
| **Factor 9:** Developmental strategies | 0.760 ** | 0.785 ** | 0.281 ** | −0.058 | 0.733 ** | 0.743 ** | 0.704 ** | 1 |

** Correlation significant at the level of 0.01 (bilateral); * Correlation significant at the level of 0.05 (bilateral).

Finally, the above dimensions and factors were equally correlated directly and positively, with all of them doing so at the level of 0.01 (Table 6). The greatest correlation strength was seen between the dimension describing learning strategies and the dimension describing resource management strategies (r = 0.862).

**Table 6.** Correlation between dimensions.

| | Dimension 1 | Dimension 2 |
|---|---|---|
| **Dimension 1: Motivation Scale** | 1 | |
| **Dimension 2: Learning Strategies** | 0.206 ** | 1 |
| **Dimension 3: Resource Management Strategies** | 0.139 ** | 0.862 ** |

** Correlation significant at the level of 0.01 (bilateral).

## 4. Discussion

To establish motivation towards learning in the university context, it is considered essential to have questionnaires that reliably measure what they purport to measure. Thus, the main objective of the present study was to analyze the psychometric properties of the MSLQ-SF and to observe its adaptation and application to the social science and health Science university population of the

University of Granada. The importance given to learning strategies will determine the cognitive resources used by students when they receive directed learning (Valle et al. 1998; Inzunza et al. 2018).

The results obtained were satisfactory in terms of the Cronbach alpha coefficients produced. This was the case for the overall scale, as well as for the eight factors (after deletion of the factor describing guidance to set intrinsic goals) and the three dimensions. This suggests that the examined instrument is both valid and reliable for estimating motivation and learning strategies as part of a multidimensional model. To better understand the present findings, the contributions of Pintrich (2004), Duncan and McKeachie (2005), and Chacón-Cuberos et al. (2018) should be consulted in relation to the effects on motivation, learning, and administration of resources by university students. When a student participates in his/her own learning and takes a perspective of self-regulated learning, he/she becomes capable of setting goals, monitoring his/her learning, and controlling his/her motivation (Kassab et al. 2015).

The motivation scale dimension maintained consistent results, with factors loading in a consistent manner. On the other hand, in the case of the learning strategies dimension, the influence of some variables was diminished, as can be seen with the administration of resource strategies variable. Despite this, the test had already been implemented extensively in the educational context with both secondary school and university students (Rao and Sachs 1999) and with Colombian students of health sciences (Sabogal et al. 2011). However, it had not been previously applied to a great extent in Spanish university populations.

As has already been mentioned, the obtained data supported the theory of learning motivation, presenting acceptable fit indices, and a good internal consistency and temporal stability, presenting values that exceeded 0.700, showing adequate reliability. It should also be noted that the data obtained suggest some issues that should be addressed in forthcoming studies.

This investigation found a high correlation between all dimensions, especially between learning strategies and resource management. We understand these results may be context specific. Similarly, the importance of motivation in the self-regulatory processes should be considered, supporting the results obtained by Ozan et al. (2012) and Sepúlveda et al. (2014), who found associations between self-regulation and learning strategies in relation to self-efficacy and academic performance. Indeed, the research team supported the contemporary vision of crafting strategies that promote the creativity, engagement, and connection aspects of student learners (Van der Wal-Maris et al. 2018; Virtanen et al. 2018).

The findings showed elaboration strategies, organization, metacognition, and critical thinking, as well as effort and study habits to be positively inter-related. Strong relationships were found in the student sample between learning and resource management. Therefore, as proposed by Effeney et al. (2013), it is probable that the correlations between cognitive-type learning strategies and the educational work done by students in their home can be used to inform strategies in their study routines and studying habits and act as essential factors of self-regulation. Evidence from this research shows that organizational strategies and time management are essential elements of academic achievement (Britton and Tesser 1991; Garavalia and Gredler 2002; Peck et al. 2018).

## 5. Conclusions

Finally, the results obtained in this study make it possible to conclude that the Motivation and Strategies Learning Questionnaire-Short Version meets the technical specifications of validity and reliability for use with university students of social and health sciences. In conclusion, the main achievement of the present research is the adaptation and validation of a measurement instrument from the MSLQ, for use in the context of teaching-learning processes.

Certainly, it would be relevant to investigate the validity of the instrument's criteria by comparing it with another that has similar characteristics. This would be useful for establishing possible differences within other variables (e.g., gender, campus, course, etc.).

Key limitations of the present study include the fact that it employed a self-report questionnaire within a relatively narrow sample of health sciences and social sciences students. A more inclusive sample may well have produced more reliable and informative findings.

**Author Contributions:** Conceptualization, F.Z.O. and R.C.C.; methodology, A.M.M.; software, J.L.U.J; validation, A.M.M., J.L.U.J., and R.C.C.; formal analysis, J.L.U.J; investigation, A.M.M.; resources, F.Z.O.; data curation, J.L.U.J; writing, original draft preparation, F.Z.O.; writing, review and editing, J.L.U.J.; visualization, X.X.; supervision, R.C.C.; project administration, R.C.C.; funding acquisition, F.Z.O.

**Funding:** This research received no external funding

**Conflicts of Interest:** The authors declare no conflict of interest.

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
