# Peer review of "Analysis of the Psychometric Properties of the Motivation and Strategies of Learning Questionnaire—Short Form (MSLQ-SF) in Spanish Higher Education Students"

_socsci, doi:10.3390/socsci8050132_

Round 1

Reviewer 1 Report

The purpose of this research is to analyze the psychometric properties of the Motivational Strategies for Learning Questionnaire short form -MSLQ-SF.  Although it is an  interesting article, the paper would be stronger if the following comments/suggestions are responded:

 p.1 -The definitions of Motivation need to be addressed in section 1.  

p.2 “Pintrich et al. (1988) used the Motivational Strategies for Learning 53 Questionnaire - MSLQ, with values of reliability 0.750; further on Pintrich. (1993) did a modification 54 of this instrument limiting it to 40 items and naming it MSLQ-SF; Roces et al. (1995) adapted it to the 55 Spanish context and they call it CEAM II (Learning and Motivation Strategies Questionnaire). “- There is no section of the literature review in this manuscript. I think it is important that the author(s) can include some literature review or theoretical background in the following aspects:

The  Motivational Strategies for Learning Questionnaire (e.g.,  the differences between the original and the short version, how it was originally designned and used in the US, and how it was used in different cultural contexts.

MSLQ dominance and dimensional structure (e.g. scales and subscales of the original version).

This paper would have been more persuasive if the author(s) had paid more attention to the section.

p. 9- “Conclusions” -- Please consider providing more details about the limitations of this study. This needs to be clarified.  

p.8-p.9 - A brief discussion is provided based on the findings. But It’s better if the author(s) can provide the discussion with more current literature.

Author Response

The authors are grateful for the assessments made by the reviewer, and have proceeded to solve them all as they are reflected in the attached file, where all the changes are commented, and in the new version where they have been put in red color and with the control of changes
In case of any new change, please inform us.
regards

Reviewer 2 Report

I enjoyed reading your manuscript. Validating an instrument is always helpful to the field.

Below are some suggestions that you might want to consider.

#1 the manuscript needs to follow up Social Sciences template, e.g., the abstract (it is not common to see (1),(2).. in this section), spacing, indention, or font (e.g., p in p-value needs to be italic) 

#2 In-text reference need to follow Social Sciences template and need to be consistent.

#3 It is too brief line 43-46. What did these studies find? You might want to strengthen your literature review.

Also, there is a new publication "Examining cross-cultural transferability of self-regulated learning model: an adaptation of the Motivated Strategies for Learning Questionnaire for Chinese adult learners Chinese adult learners" in Educational Studies that you might want to refer. They are doing a very similar thing like you.  Their literature review section might help you to enrich yours.

#4 As for reporting KMO, RMSEA, Cronbach Alpha, and other indexes, you might want to refer to some threshold, then readers can understand how good the results are.

#5 You might also need to specify which method you conducted extraction, principal components or principal axis, or others, as well as the methods for rotation, varimax, direct oblimin or others.

#6 You might want to strengthen your conclusion part and make more connections with your introduction/literature review section. 

Author Response

(The authors gave the same response as above.)

Round 2

Reviewer 2 Report

Dear Authors,

Thank you for editing the manuscript.

It seems like points #4-#6 are not addressed, although in the response letter it said the issues were fixed. Is it possible that this version is not the most updated one? Or there are comments that I can not see due to the PDF format.

You can specify the line numbers where you added the comments/edits.

The conclusion section needs to be expanded.

You might need to smoothe the transition between conclusion and limitation.

Thank you!

Author Response

The authors appreciate the contributions made by this reviewer and that has surely strengthened the document, the first of them we consider has been corrected, the second one has tried to change but we do not know if it is that way. The authors indicate the possibility of combining discussion and conclusions.

Round 3

Reviewer 2 Report

It is improved. Thank you!